# Impact of Cognitive Disturbances and Clinical Symptoms on Disability in Patients with Paranoid Schizophrenia: A Study of a Bulgarian Clinical Sample

**DOI:** 10.3390/ijerph20032459

**Published:** 2023-01-30

**Authors:** Ivanka Veleva, Kaloyan Stoychev, Maya Stoimenova-Popova, Eleonora Mineva-Dimitrova

**Affiliations:** 1Department of Psychiatry and Medical Psychology, Faculty of Public Health, Medical University, 5800 Pleven, Bulgaria; 2Department of Social Medicine and Health Management, Faculty of Public Health, Medical University, 5800 Pleven, Bulgaria

**Keywords:** schizophrenia, disability, WHO-DAS 2.0, positive, negative, cognitive symptoms, psychosocial functioning

## Abstract

The study aimed to assess the impact of clinical symptoms and cognitive impairment on disability in patients with paranoid schizophrenia (PS). Methods: 108 patients with schizophrenia were included (66 male and 42 female). Their average age was 38.86 ± 10.02 years and the disease duration was 12.80 ± 8.20 years, with mean disease onset of 24 years. Clinical symptoms were assessed with the PANSS, and cognitive performance was measured using a seven-item neurocognitive battery. The disability level of the subjects was assessed using the World Health Organization—Disability Assessment Schedule 2.0 (WHO-DAS 2.0). The relation between the variables studied was assessed using Spearman’s rank correlation coefficient (*r_s_*) at a probability level of *p* < 0.05. Results: An increase in symptom severity resulted in worsening of the “participation in society” (*r* = 0.56, *p* < 0.01), “life activities—household” (*r* = 0.55, *p* < 0.01), and “getting along with people” (*r* = 0.59, *p* < 0.01) WHO-DAS 2.0 domains. Positive symptoms (13.89 ± 3.48) correlated strongly with “getting along with people” (*r* = 0.55, *p* < 0.01), “life activities—household” (*r* = 0.58, *p* < 0.01), and “participation in society” (*r* = 0.62, *p* < 0.01), and negative symptoms (14.25 ± 4.16) with “participation in society” (*r* = 0.53, *p* < 0.01) and “life activities—household” (*r* = 0.48, *p* < 0.01). Symptoms of disorganization (15.67 ± 4.16) had the highest impact on “life activities—household” (*r* = 0.81, *p* < 0.01), “getting along with people” (*r* = 0.56, *p* < 0.05), and “participation in society” (*r* = 0.65, *p* < 0.01). Episodic memory (*r* = −0.28, *p* < 0.01) was remotely related to comprehension and communication. The information processing speed (*r_s_* = 0.38, *p* < 0.01), visual memory (*r_s_* = −0.30, *p* < 0.01), and focused executive functions showed moderate correlations with all domains on the WHO-DAS 2.0 scale (*r_s_* = 0.38, *p* < 0.01). Attention (*r_s_* = −0.33, *p* < 0.01) was moderately related to community activities. Semantic (*r_s_* = −0.29, *p* < 0.01) and literal (*r_s_* = −0.27, *p* < 0.01) verbal fluency demonstrated weak correlations with “cognition—understanding”, “getting along with people”, and “participation in society”. Conclusion: Symptoms of disorganization and disturbed executive functions contribute most to disability in patients with schizophrenia through impairment of real-world functioning, especially in social interactions and communication. Severe clinical symptoms (negative and disorganization-related ones) as well as deficits in executive function, verbal memory, and verbal fluency cause the biggest problems in the functional domains of interaction with other people and participation in society.

## 1. Introduction

Schizophrenia is a severe mental illness that, despite its relatively low prevalence of approximately 1% [1,2], is ranked as the 12th most disabling disorder among 310 diseases and injuries worldwide. Starting in early adulthood and having a chronic course with incomplete remissions in most cases, schizophrenia causes enormous social burden and is, therefore, a leading public health problem. In general, 70–90% of individuals with schizophrenia have occupational and housing problems [3,4,5,6] and 50% have no social contacts [7,8]. Schizophrenic patients live an average of 20 years less than the general population [9,10].

Socio-occupational decline in schizophrenia patients is mostly due to negative [11] and disorganization-related symptoms [12] as well as to cognitive deficits [13]. Negative symptoms tend to precede positive ones and significantly predict impairment in community activities and communication as well as poor work/study engagement [14]. Disorganization symptoms on the other hand are more closely related to daily functioning impairments [12]. While the adverse functional impact of negative symptoms is mostly due to the diminished/lacking overall motivation as well as loss of interest in social interactions, disorganization symptoms seem to overlap with cognitive dysfunction and particularly with symptoms such as difficulties in abstract thinking and poor attention [15]. A strong capacity for abstract thinking and a good attention span, along with other cognitive abilities (primarily executive functions), are essential for occupational and social functioning, which require maintaining perceived stimuli and monitoring, prioritizing, and manipulating them to plan and achieve the appropriate response. With respect to cognitive symptoms, the following relations have been established: the impairment of executive functions and attention is associated with poor overall functioning [16], the reduced cognitive flexibility affects quality of life and interpersonal relationships [17], while disturbances in verbal memory and speed of information processing/attention are predictors of poor social functioning [18].

Based on the limited research in this area in Bulgaria, the present study aims to assess the impact of clinical symptoms and cognitive impairment on disability in a clinical sample of patients with paranoid schizophrenia.

## 2. Materials and Methods

We examined 108 patients with paranoid schizophrenia (PS) (66 male and 42 female) with an average age of 38.86 ± 10.02 years, an average duration of the disease 12.80 ± 8.20 years, and an average age of onset of the disease of 24.00 years (Med 25.0; Moda 22.0) treated during the period 2017–2019 at the Psychiatry Department of Pleven University Hospital, Bulgaria. Participants fulfilled the following inclusion criteria: (1) ICD-10 [19] PS-based diagnosis; (2) a second-generation antipsychotic current therapy that had produced a stable state defined by a lack of serious relapse (a change of ≥25% of the total PANSS score) in the three months before the study entry; (3) presence of clinically significant positive, negative, and/or disorganization symptoms determined by a score of at least “mild” on at least two of the corresponding PANSS items according to the van der Gaag et al.’s five factor model [20]. The eliminating criteria were as follows: (1) lifetime history of another psychiatric state or history of substance use disorder(s) in the last year; (2) any neurological condition and/or physical illness/impairment that interferes with the proper execution of study procedures.

The subjects’ disability level was assessed using the full version of World Health Organization—Disability Assessment Schedule 2 (WHO–DAS 2.0) [21]. It contains 36 items in 6 domains: Domain 1 (Cognition—understanding and communicating); Domain 2 (Mobility—moving and getting around); Domain 3 (Self-care); Domain 4 (Getting along with people); Domain 5 (Life activities—household, school/work); Domain 6 (Participation in society). Standardized summary scores range from 0 to 100, with higher scores indicating greater levels of disability in functioning [21].

The following neurocognitive battery was used in the sequence indicated:

Ten words Luria test: a verbal memory (VM) test that measures fixation (immediate recall), retention, and reproduction (delayed recall).

Trail making yest parts A and B (TMT A&B): a standardized assessment test of processing speed, attention switching, cognitive flexibility, distributed and selective attention, ability to form concepts, and executive functions (part B only).

Logical memory test: measures both episodic memory and disorders of thinking.

Digit symbol substitution test (DSST): assesses visuo-perceptual functions (including scanning), attention, executive functions (including working memory), and motor speed.

Verbal fluency test (VF): measures executive functions and semantic memory.

Benton visual retention test (BVRT): designed to measure visual perception, visual memory as well as basic visual abilities, visuo-motor coordination, and perceptual-motor integration.

A detailed description of the abovementioned neurocognitive tools can be found elsewhere [22] along with the test results’ interpretation according to the normative performance values globally and for the Bulgarian population.

All patients were capable of providing informed consent for participation, which they all did before their enrolment in the study. The study itself was approved by the Research Ethics Committee of Medical University of Pleven. All participants were native Bulgarian language speakers having at least eight years of education. Additionally, all of them were right-handed according to the Edinburgh Handedness Inventory—Short Form [23].

The exploration of the relation between social functioning, severity of clinical symptoms, and neurocognitive test battery performance was carried out by means of correlation analysis for linear dependence—Pearson correlation coefficient (*R*) and Spearman’s rank correlation coefficient (*r_s_*) at probability *p* < 0.05. All WHO-DAS 2.0 results were calculated using the complex IRT (item–response–theory) method. Multiple regression analysis was conducted to examine the relationship of symptomatology and cognitive impairment with individual domains of WHO-DAS 2.0. Our primary task was to (1) determine whether a relationship between different symptom groups (positive, negative, and of disorganization) and cognitive impairment on the one hand and WHO-DAS 2.0 domains on the other objectively exists and (2) to measure the specific quantitative relationships by assessing the influence of the factor variables on the outcome variable.

Data processing was performed using IBM SPSS v.26.

## 3. Results

The average severity of the disease based on PANSS score was 71.80 ± 5.10. The increase of symptom severity resulted in worsening of “participation in society” (*r* = 0.56, *p* < 0.01), “life activities—household” (*r* = 0.55, *p <* 0.01), and “getting along with people” (*r* = 0.54, *p* < 0.01). Positive symptoms (13.89 ± 3.48) correlated higher with “getting along with people” (*r* = 0.55, *p* < 0.01), “life activities—household” (*r* = 0.58, *p* < 0.01), and “participation in society” (*r* = 0.62, *p* < 0.01), and negative symptoms (14.25 ± 4.16) with “participation in society” (*r* = 0.53, *p* < 0.01) and “life activities—household” (*r* = 0.48, *p* < 0.01). Symptoms of disorganization (15.67 ± 4.16) had the highest impact on “life activities—household” (*r* = 0.81, *p* < 0.01), “getting along with people” (*r* = 0.56, *p* < 0.05), and “participation in society” (*r* = 0.65, *p* < 0.01). All correlations were significant at *p* = 0.01. “Getting around” had no relation with disease severity.

Significant differences were found between groups by education level and occupation (employed/students, unemployed, and disabled)—χ^2^ = 9.724, df = 4, *p* = 0.045, Cramer’s V = 0.212.

Fixation on auditory verbal memory (10 words Luria test) had weak correlations with all domains of the WHO-DAS 2.0 scale (*r* = −0.28, *p* = 0.003), excluding “getting around”. The retention of verbal material (verbal memory retention sub-component of 10 words Luria test) had weak correlations with domains “self-care” (*r* = −0.20, *p* = 0.036), “getting along with people” (*r* = −0.21, *p* = 0.033), “life-activities—household” (*r* = −0.24, *p* = 0.011), and “participation in society” (*r* = −0.24, *p* = 0.013). Reproduction (10 words Luria) correlated slightly only with the “self-care” domain (*r* = −0.29, *p* = 0.002). Verbal fluency—semantic (*r_s_* = −0.29, *p* = 0.002) and literal (*r_s_* = −0.27, *p* = 0.005)—also showed slight correlation with “cognition—understanding”, “self-care”, “getting along with people”, “life activities—household”, and “participation in society”. Psychomotor speed (measured using TMT-A) (*r_s_* = 0.32, *p* = 0.001), focused attention (assessed using DSST) (*r_s_* = −0.33, *p* = 0.001), and executive functions (graded using TMT-B) (*r_s_* = 0.38, *p* = 0.000) showed moderate correlations with all domains of the WHO-DAS 2.0. Moderately pronounced correlations were found with the number of wrong answers in visual memory (*r_s_* = −0.30, *p* = 0.001), measured using BVRT, and weak correlations with the number of correct answers (*r_s_* = −0.28, *p* = 0.004).

The correlation analysis revealed a significant positive correlation between negative symptoms and “life activities—household” (*r* = 0.48, *p* < 0.01) and “participation in society” (*r* = 0.53, *p* < 0.01). A positive correlation was found between executive functions and “life activities—household” (*r* = 0.30, *p* < 0.05) and “participation in society” (*r* = 0.27, *p* < 0.01). Further, regression analysis (Tabl. 8) revealed that negative symptoms and executive functions significantly influence “life activities—household” with 65.0% coefficient of determination (R^2^ = 0.65, *p* < 0.01) and “participation in society” with 60.0% coefficient of determination (R^2^ = 0.60, *p* < 0.01).

We observed a significant positive correlation between disorganized symptoms and “cognition—understanding and communicating life” (*r* = 0.49, *p* < 0.01), “life activities—household” (*r* = 0.81, *p* < 0.01), and “participation in society” (*r* = 0.65, *p* < 0.01). Regression analysis (Tabl. 8) revealed that disorganized symptoms and verbal fluency, may influence “cognition—understanding and communicating life” with 56.0% coefficient of determination (R^2^ = 0.56, *p* < 0.01). Disorganized symptoms alone influenced “life activities—household” with a 63.0% coefficient of determination (R^2^ = 0.63, *p* < 0.001) and “participation in society” with a 58.0% coefficient of determination (R^2^ = 0.58, *p* < 0.01).

## 4. Discussion

The current study explores the impact of cognitive disturbances and clinical symptoms on disability in a sample of hospitalized paranoid schizophrenia subjects. The prevalence of male patients (see Table 1) in our sample is in line with the literature data stating that due to the earlier onset and the more severe disease course in males, the latter tend to be overrepresented in society and in hospital samples, respectively [24]. Our results show that social functioning is slightly affected by positive symptoms of schizophrenia (Table 2). This finding is consistent with the results of several meta-analyses [12,25,26,27] showing that the relation between positive symptoms and functioning in community is insignificant. Therefore, symptoms such as hallucinations and delusions may not be a severe obstacle to socialization and employment in subjects with schizophrenia. A probable explanation for this could be that, with time, patients develop various ways to compensate for these symptoms and/or to ignore them in social interactions or at work. This suggests that focusing mainly on symptoms such as hallucinations and thought disorders, as traditional treatment approaches do, will not necessarily result in better functioning.

We found significant correlations between negative symptoms and the performance of daily activities (WHO DAS 2.0 item D_5_) as well as the relationship with others (item D_4_)_._ In addition to that, we observed a strong correlation with social participation D_6_ (Table 2), which is in line with previous reports [27,28]. These findings are not surprising given the fundamental predictive role of negative symptoms in the overall functional outcome in schizophrenia [14]. Being more proximal to the neurobiological perturbations that drive the disease and hence occurring earlier in its course, negative symptoms significantly contribute to the impairment of social activities and communication deficits. Inside the group of negative symptoms, anhedonia and apathy may particularly inhibit schizophrenic patients’ motivation and social skills. As the available antipsychotic treatments rarely, if ever, affect negative symptoms, they surely do not guarantee improvement in daily functioning outcomes [29].

Disorganization symptoms in our study were more closely related to “getting along with people” and “participation in society” (Table 2). In this respect, our results support previous research [30,31] that shows a positive correlation between disorganization symptoms and difficulties in everyday functioning and social performance (interpersonal relationships and social activities). Disorganization symptoms, which like negative ones reflect an underlying dimension that is close to the core of the illness, are inversely associated with long-term functioning [31,32]. Ventura et al. (2010) [15] assume that this association may be due to the adverse impact on communication and social interactions caused by disorganization symptoms (e.g., disorganized speech interferes with communication) which unlike positive ones, cannot be overcome by compensatory mechanisms [12]

In the course of the study, we found a relation between cognitive performance and social functioning in patients with PS. Poor or absent occupational functioning in individuals with schizophrenia is the result of many various factors related to the illness, some of which predate its onset. Two of the most significant among these are poor premorbid functioning and low levels of education. In our study, patients with a higher educational status showed lower impairment in all WHO DAS 2.0 domains. Lower educational status puts people at a disadvantage in the labour market, and this is valid for the general population as well as for schizophrenia patients. In this way, both factors—impaired premorbid functioning and low education—are an obstacle to professional realization in people who subsequently become ill with schizophrenia.

Global psychosocial functioning, including work/study engagement, is predicted by negative symptoms and attention [11]. Our data support this assumption and show that impaired attention worsens domain 5 of the WHO-DAS (life activities). In our study, focused attention/interference control was associated more with community activities, as well as with the two tests indicative for information processing speed—DSST (Table 3) and TMT-A (Table 4). Therefore, it could be hypostatized that attention influences everyday functioning both independently and through its influence on information processing speed.

Out results show that verbal memory (Table 5) and verbal fluency (Table 6) are closely related to understanding and communication (D_1_). Verbal memory especially significantly determines the impairment degree in community activities, communication, and interpersonal relationships. Verbal fluency deficit in its turn also causes difficulties in communication, which hampers establishing and keeping friendships [33]. Individually, verbal memory, focused attention, and information processing speed have moderate effects on functioning, but their combined effects are significant. The cognitive subdomains within these individual domains (e.g., learning, attention, executive functions) appear to be less related to the impairment than the global neurocognition. The domains have small to moderate correlations with functional outcome in schizophrenia, while the combined results show moderate to large correlations with impairment. Difficulty engaging in a task/work may be a key impairment in patients with schizophrenia. The information processing speed (Table 3 and Table 4) and visual memory (Table 7) are more related to community activities (D_6_). In their review of the neural correlates of the perception of socio-emotional stimuli, Green & Leitman (2008) [34] raise the hypothesis that the perception deviations and integration of visual information may be at the basis of impaired social functioning. Thus, disturbances in visual perception in schizophrenia appear to be a basic component of social maladaptation. Misinterpretation of visual material distorts the perception of socially relevant stimuli necessary for adequate interactions. It is believed that these violations can be determined by the features of the so-called “mirror neurons” in the parietal cortex responsible for recognizing the mental state of other people [34].

Self-care (D_3_) and daily living activities (D_5_) are basic and are the last to be lost with advancing age or in the course of illness, because they are built first in the course of individual development. We found that mobility (D_2_) was slightly correlated with psychomotor speed measured using TMT-A and DSST, but not using TMT-B (see Table 8).

A particular strength of our study is that it is one of the very few that measure schizophrenia-related disability via WHO-DAS 2.0—an instrument recommended by DSM 5. Alongside kthat, several important limitations must be pointed out. The first one is its cross-sectional design. A longitudinal design could better clarify how changes in clinical symptoms influence psychosocial functioning and disability. Second, the influence on disability of factors associated with the disease (family history of mental illness, age of disease onset, illness duration, etc.) are not taken into account. In addition, in our analysis, we did not take into account the influence of protective factors such as availability of a partner, higher income, or stronger social support.

## 5. Conclusions

Poor occupational functioning is a hallmark of schizophrenia contributing to the burden of disease on individuals, their families, and society. WHODAS 2.0 is a clinically useful tool for the assessment of psychosocial functioning in patients with schizophrenia. Symptoms of disorganization and disturbed executive functions contribute the most to disability in patients with schizophrenia through impairment of real-world functioning, especially in social interactions and communication. Severe clinical symptoms (negative and disorganization-related ones) as well as deficits in executive function, verbal memory, and verbal fluency cause the biggest problems in the functional domains of interaction with other people and participation in society.

## Figures and Tables

**Table 1 ijerph-20-02459-t001:** Sociodemographic and clinical characteristic of PS patients.

**Characteristic**	*n* = 108
Gender *(n, %)* χ^2^ = 5.333, df = 1, Asymp. Sig0.021 - Female - Males	42 (38.9%) 66 (61.1%)
Age (Mean ± SD)	38.86 *±* 10.020
Education *(n, %)* - College/University degree - High school - Elementary school	31 (28.7%) 53 (49.1%) 24 (22.2%)
Employment status *(n, %)* - Employed - Unemployed - Retired/disabled	41 (38.0%) 15 (13.9%) 52 (48.1%)
Family history for schizophrenia *(n, %)* - Family history positive - Family history negative	52 (48.1%) 56 (51.9%)
Mean duration of schizophrenia (Mean ± SD)	12.68 *±* 8.202
Mental illness onset (Me; IQR)	25; 10
PANSS (Mean ± SD) - Total - Positive ^1^ - Negative ^2^ - Disorganization ^3^	71.44 ± 7.50 13.89 *±* 3.48 14.25 *±* 4.16 15.67 *±* 4.16

^1^ Positive sub-score of the PANSS includes items P1, P3, P5, P6, G1, G9. ^2^ Negative sub-score of the PANSS includes items N1, N2, N3, N4, N6, G7, G16. ^3^ Disorganization sub-score of the PANSS includes items P2, N5, N7, G5, G10, G11, G12, G13, G15.

**Table 2 ijerph-20-02459-t002:** Correlation with PANSS.

**Domain**	PANSS			
	Total Score	Positive Symptoms	Negative Symptoms	Disorganized Symptoms
Domain 1 (Cognition—understanding and communicating)	0.39 **	0.41 **	0.34 **	0.49 **
Domain 2 (Mobility—moving and getting around)	NS	NS	0.21 *	NS
Domain 3 (Self-care)	0.47 **	0.46 **	0.40 **	0.47 **
Domain 4 (Getting along with people)	0.59 **	0.55 **	0.39 **	0.56 **
Domain 5 (Life activities—household)	0.55 **	0.58 **	0.48 **	0.81 **
Domain 6 (Participation in society)	0.56 **	0.62 **	0.53 **	0.65 **

Pearson correlation coefficient (R). * Correlation is significant at the 0.05 level (2-tailed). ** Correlation is significant at the 0.01 level (2-tailed).

**Table 3 ijerph-20-02459-t003:** Correlations with DSST.

Domain	*r_s_*	Sig. (2-Tailed)
Domain 1 (Cognition—understanding and communicating))	−0.31	0.001
Domain 2 (Mobility—moving and getting around)	−0.23	0.015
Domain 3 (Self-care)	−0.28	0.003
Domain 4 (Getting along with people))	−0.31	0.001
Domain 5 (Life activities—household)	−0.29	0.003
Domain 6 (Participation in society)	−0.31	0.001

Spearman correlation coefficient (*r_s_*).

**Table 4 ijerph-20-02459-t004:** Correlations with trail making test (TMT).

**Domain**	Trail Making Tests (In Seconds)	
	**Part A**	**Part B**
Domain 1 (Cognition—understanding and communicating)	0.30 **	0.28 **
Domain 2 (Mobility—moving and getting around)	0.20 *	NS
Domain 3 (Self-care)	0.33 **	0.23 *
Domain 4 (Getting along with people)	0.35 **	0.33 **
Domain 5 (Life activities—household)	0.34 **	0.30 **
Domain 6 (Participation in society)	0.36 **	0.27 **
Total	0.38 **	0.32 **

Spearman correlation coefficient (*r_s_*). * Correlation is significant at the 0.05 level (2-tailed). ** Correlation is significant at the 0.01 level (2-tailed).

**Table 5 ijerph-20-02459-t005:** Correlation with Luria 10 words.

**Domain**	Luria 10 Words		
	**Fixation Success Rate (S_mf_)**	**Retention Success Rate (S_ret_)**	**Reproduction Success Rate (S_rep_)**
Domain 1 (Cognition—understanding and communicating)	−0.29 **	NS	NS
Domain 2 (Mobility—moving and getting around)	NS	NS	NS
Domain 3 (Self-care)	NS	−0.20 *	−0.29 **
Domain 4 (Getting along with people)	−0.29 **	−0.21 *	NS
Domain 5 (Life activities—household)	−0.29 **	−0.24 *	NS
Domain 6 (Participation in society)	−0.25 **	−0.24 *	NS
Total	−0.28 **	−0.25 **	NS

Pearson correlation coefficient (R). * Correlation is significant at the 0.05 level (2-tailed). ** Correlation is significant at the 0.01 level (2-tailed).

**Table 6 ijerph-20-02459-t006:** Correlations with verbal fluency.

**Domain**	Verbal Fluency (VF)	
	Semantic VF (VF_s_)	Literal VF (VF_l_)
Domain 1 (Cognition—understanding and communicating)	−0.30 **	−0.26 **
Domain 2 (Mobility—moving and getting around)	NS	NS
Domain 3 (Self-care)	−0.27 **	−0.21 *
Domain 4 (Getting along with people)	−0.31 **	−0.22 *
Domain 5 (Life activities—household)	−0.27 **	−0.26 **
Domain 6 (Participation in society)	−0.24 *	−0.26 **
Total	−0.29 **	−0.27 **

Spearman correlation coefficient (*r_s_*). * Correlation is significant at the 0.05 level (2-tailed). ** Correlation is significant at the 0.01 level (2-tailed).

**Table 7 ijerph-20-02459-t007:** Correlations with Benton visual retention test (BVRT).

**Domain**	BVRT	
	Correct	**Error**
Domain 1 (Cognition—understanding and communicating)	−0.31 **	0.32 **
Domain 2 (Mobility—moving and getting around)	NS	NS
Domain 3 (Self-care)	−0.22 *	0.22 *
Domain 4 (Getting along with people)	−0.27 **	0.24 *
Domain 5 (Life activities—household)	−0.26 **	0.31 **
Domain 6 (Participation in society)	−0.24 *	0.28 **
Total	−0.28 **	0.30 **

Spearman correlation coefficient (*r_s_*). * Correlation is significant at the 0.05 level (2-tailed). ** Correlation is significant at the 0.01 level (2-tailed).

**Table 8 ijerph-20-02459-t008:** Influence of clinical symptoms (positive, negative and disorganization symptoms) and cognitive impairment on WHO DAS 2.0 domains—Multiple regression analysis model summary.

Domain	R	R^2^	Adjusted R	S. E. of the Estimate	F	Sig. F
Negative Symptoms and cognitive impairment						
Domain 5	0.88	0.77	0.65	15.648	6.34	0.000
Domain 6	0.86	0.74	0.60	12.329	5.33	0.001
Positive Symptoms and cognitive impairment						
Domain 5	0.82	0.68	0.52	18.323	4.19	0.002
Domain 6	0.83	0.69	0.54	13.347	4.45	0.001
Disorganized Symptoms and cognitive impairment						
Domain 1	0.84	0.70	0.56	11.561	4.74	0.001
Domain 3	0.77	0.58	0.37	13.769	2.79	0.019
Domain 5	0.87	0.76	0.63	15.920	6.20	0.000
Domain 6	0.85	0.72	0.58	12.652	5.18	0.001

## Data Availability

Data supporting the reported results can be found at the Department of Psychiatry and Medical Psychology of Pleven Medical University (Ivanka Veleva: ivanka.sirashky@gmail.com).

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
