# Peer review of "Impact of Cognitive Disturbances and Clinical Symptoms on Disability in Patients with Paranoid Schizophrenia: A Study of a Bulgarian Clinical Sample"

_ijerph, 2023, doi:10.3390/ijerph20032459_

Round 1

Reviewer 1 Report

The authors have managed to achieve a very good coherence between the title,
summary, substantive part
s, and the conclusion of the presented article.
After the plagiarism check, was established 100% authenticity.

Тhe authors were able to make a detailed review of the scientific literature on this topic.
Presented article has the necessary qualities for originality, scientificity and reliability of
the presented data.
The authors are well acquainted with the topic, have an innovative approach, and
defend their theses well.
It would be good in similar studies to include a patient group with combined therapy.
It is not clear what is the patients' social support system.
It would be interesting to have a comparison with another group of patients,
who received psychosocial therapy, rehabilitation and resocialization together
with antipsychotic treatment.

Author Response

Comment: The authors have managed to achieve a very good coherence between the title, summary, substantive parts, and the conclusion of the presented article.

After the plagiarism check, was established 100% authenticity.

Тhe authors were able to make a detailed review of the scientific literature on this topic.

Presented article has the necessary qualities for originality, scientificity and reliability of the presented data.

The authors are well acquainted with the topic, have an innovative approach, and defend their theses well.

Reply: Thank you very much for your most useful comments and recommendations.

Comment: It would be good in similar studies to include a patient group with combined therapy.

Reply: As for the first comment – that it will good in similar studies to include a patient group with combined therapy – we totally agree but we do not have at this point access to such a group of patients.

Comment: It is not clear what is the patients' social support system.

Reply: As for the second comment – that it is not clear what is the patients’ support system – we also agree, but as we have not collected data in this aspect. We have pointed this as one of the limitations of our study.

Comment: It would be interesting to have a comparison with another group of patients, who received psychosocial therapy, rehabilitation and resocialization together with antipsychotic treatment.

Reply: For the third comment - that a comparison with a group receiving some kind of psychosocial therapy alongside their pharmacotherapy will be useful and informative - again we totally agree with the author but so far, we do not have access to such a sample of patient.

We accept them but unfortunately in our dataset there is no data on the quality of social support system of the patients, i.e.

Similarly, due to absence of data, we are not able to compare the present group of patients with another group(s) who have received or are receiving some sort of psychosocial therapy.

You can find the corrected manuscript attached.

We remain available if further corrections and/or clarifications are needed.

Kind Regard,

The authors

Reviewer 2 Report

Dear editor,

Plese find below the my review for the manuscript titled with “Impact of Cognitive Disturbances and Clinical Symptoms on Disability in Patients with Paranoid Schizophrenia: A Study of 3 Bulgarian Clinical Sample”.

The authors investigated the effects of cognitive disturbances and clinical picture on disability in patients with paranoid schizophrenia. The authors evaluated the measurement results they obtained by performing correlation analyzes in various combinations. In this evaluation process, it is thought that some confounding effects may have been overlooked. For example, both cognitive factors and positive and negative findings of schizophrenia may be associated with disability. Therefore, it would be appropriate to perform a regression analysis or to consider confounding factors together with individual correlation analyses.

In addition to these, other data are needed in addition to the demographic information of the patients so that the reader can better interpret the presented data. For example, in the evaluation of disability, it would be appropriate to present data such as the drugs used by the patients and their doses, the number of treatment-resistant cases, and the rate of patients receiving combination therapy. It would even be advantageous to use them in statistical comparisons.

It would be appropriate to state why 108 patients were included in the study, whether a power analysis was performed, and whether a rule was set in determining gender distribution.

The titles of Tables 2-7 do not fully reflect the content; they should be written clearly, allowing a reader to interpret the table without looking at the article content. Table 5 does not have a column heading. On the other hand, it would be appropriate to give the long forms of all abbreviations under the table.

Various times in the article, 2, 3 and 4 digits are included in decimal writing. It will be appropriate to write them in accordance with the standard in the rules of the journal and in a way that will be the same everywhere.

In the "conclusion" section, both in the abstract and in the text, the authors reported a result that they did not actually studied, and interpreted somewhat arbitrarily. It would be appropriate for them to make this interpretation a little more cautiously and in a way that only conveys the results they have found.

Minor Suggestions,

It would be appropriate to correct the "psychiatric state" in line 86 as a psychiatric disorder.

Line in Table 1 with “Retired/disabEled”, letter E should be lower case.

Author Response

Comment: The authors investigated the effects of cognitive disturbances and clinical picture on disability in patients with paranoid schizophrenia. The authors evaluated the measurement results they obtained by performing correlation analyzes in various combinations. In this evaluation process, it is thought that some confounding effects may have been overlooked. For example, both cognitive factors and positive and negative findings of schizophrenia may be associated with disability. Therefore, it would be appropriate to perform a regression analysis or to consider confounding factors together with individual correlation analyses.

Reply: Thank you very much for your most useful comments and recommendations. We have tried to address as many of them as possible. More specifically, multiple regression analysis was conducted as per your recommendation to examine the relationship of symptomatology and cognitive impairment to individual domains of WHO-DAS 2.0.

Comment: In addition to these, other data are needed in addition to the demographic information of the patients so that the reader can better interpret the presented data. For example, in the evaluation of disability, it would be appropriate to present data such as the drugs used by the patients and their doses, the number of treatment-resistant cases, and the rate of patients receiving combination therapy. It would even be advantageous to use them in statistical comparisons.

Reply: Unfortunately, other recommendations such as taking into account the drugs taken by subjects, their doses and treatment resistant cases could not be explored as this information has not been included in our dataset. However, it is possible to add it subsequently and reanalyse data this opening up the possibility for a new publication in the future.

Comment: It would be appropriate to state why 108 patients were included in the study, whether a power analysis was performed, and whether a rule was set in determining gender distribution.

Reply: As for your comments regarding the sample – why is this particular number of patients included, has some kind of rule been set for gender distribution followed and weather a power analysis has been performed - we have only elaborated shortly in the discussion part on the overprevalence of men in our sample. We have not followed particular rule to determine gender distribution and we have not performed power analysis.

Comments: The titles of Tables 2-7 do not fully reflect the content; they should be written clearly, allowing a reader to interpret the table without looking at the article content. Table 5 does not have a column heading. On the other hand, it would be appropriate to give the long forms of all abbreviations under the table.

Various times in the article, 2, 3 and 4 digits are included in decimal writing. It will be appropriate to write them in accordance with the standard in the rules of the journal and in a way that will be the same everywhere.

Minor Suggestions,

It would be appropriate to correct the "psychiatric state" in line 86 as a psychiatric disorder.

Line in Table 1 with “Retired/disabEled”, letter E should be lower case.

Reply: Further down the list of recommendations of yours we have edited all tables and also have formatted the decimal writings according to the journal’s standards.

Comment: In the "conclusion" section, both in the abstract and in the text, the authors reported a result that they did not actually studied, and interpreted somewhat arbitrarily. It would be appropriate for them to make this interpretation a little more cautiously and in a way that only conveys the results they have found.

Reply: Finally, we have corrected the text in the conclusion section to exclude statements that do not rest on our actual research in this particular study.  

We remain available if further corrections and/or clarifications are needed.

Kind Regard,

The authors
